# A multicenter proof-of-concept study on deep learning-based intraoperative discrimination of primary central nervous system lymphoma

Xinke Zhang [1,7], Zihan Zhao[1,7], Ruixuan Wang [2,7], Haohua Chen[1,7], Xueyi Zheng [1], Lili Liu[1], Lilong Lan[1], Peng Li[1], Shuyang Wu[1], Qinghua Cao[3], Rongzhen Luo[1], Wanming Hu [1], Shanshan lyu[4], Zhengyu Zhang[5], Dan Xie [1] ✉, Yaping Ye [5] ✉, Yu Wang [6] ✉ & Muyan Cai [1] ✉

Accurate intraoperative differentiation of primary central nervous system lymphoma (PCNSL) remains pivotal in guiding neurosurgical decisions. However, distinguishing PCNSL from other lesions, notably glioma, through frozen sections challenges pathologists. Here we sought to develop and validate a deep learning model capable of precisely distinguishing PCNSL from non-PCNSL lesions, especially glioma, using hematoxylin and eosin (H&E)-stained frozen whole-slide images. Also, we compared its performance against pathologists of varying expertise. Additionally, a human-machine fusion approach integrated both model and pathologic diagnostics. In external cohorts, LGNet achieved AUROCs of 0.965 and 0.972 in distinguishing PCNSL from glioma and AUROCs of 0.981 and 0.993 in differentiating PCNSL from non-PCNSL lesions. Outperforming several pathologists, LGNet significantly improved diagnostic performance, further augmented to some extent by fusion approach. LGNet's proficiency in frozen section analysis and its synergy with pathologists indicate its valuable role in intraoperative diagnosis, particularly in discriminating PCNSL from glioma, alongside other lesions.

Accurate intraoperative diagnosis is crucial for decision-making during tumor surgery. However, differentiating between diverse primary central nervous system (CNS) tumors, including primary CNS lymphoma (PCNSL) and non-PCNSL entities like glioma, metastatic cancer, and other brain lesions, has always posed significant challenges[1,2]. PCNSL and glioma, being among the most prevalent primary brain malignancies encountered during surgeries, demand accurate and timely diagnosis due to the substantial divergence in intraoperative treatment options for these tumors within the realm of neuro-oncology[3–7].

While specific histomorphology features can aid in differential diagnosis[3–7], pathologists encounter challenges in distinguishing between diverse brain tumors, including PCNSL and non-PCNSL entities like glioma, based on hematoxylin and eosin (H&E)-stained frozen

[1]Department of Pathology, State Key Laboratory of Oncology in South China, Guangdong Provincial Clinical Research Center for Cancer, Sun Yat-sen University Cancer Center, Guangzhou 510060, China. [2]School of Computer Science and Engineering, Sun Yat-sen University, Guangzhou 510006, China. [3]Department of Pathology, The First Affiliated Hospital, Sun Yat-sen University, Guangzhou 510080, China. [4]Department of Pathology, Guangdong Provincial People's Hospital, Guangzhou 510080, China. [5]Department of Pathology, Nanfang Hospital, Soutern Medical University, Guangzhou 510515, China. [6]Department of Pathology, Zhujiang Hospital, Soutern Medical University, Guangzhou 510280, China. [7]These authors contributed equally: Xinke Zhang, Zihan Zhao, Ruixuan Wang, Haohua Chen. ✉e-mail: xiedan@sysucc.org.cn; yeyp1980@126.com; doctorwylh@163.com; caimy@sysucc.org.cn

sections in previous large-scale studies[8–10]. Neuroimaging provides valuable insights into the distinction between these tumors; however, it lacks the precision to differentiate them accurately[11]. Despite the promising results of radiomics-based machine learning approaches in discerning PCNSL and glioma, some models still exhibit room for improvement in performance, as highlighted in pooled analyses[12]. Therefore, given the intraoperative histopathological diagnosis as the gold standard for brain tumors, accurate differentiation of PCNSL remains crucial. However, pathologists lack access to immunohistochemical and molecular assays that could assist in differential diagnosis, relying mainly on interpreting cytologic and histomorphological characteristics from H&E frozen slides. Time constraints during intraoperative diagnosis place substantial pressure on pathologists to expedite diagnoses[13], leading to ambiguous findings in a notable percentage of cases (approximately 10–20%), significantly influencing

neurosurgeons' decision-making[1,2]. Hence, an urgent need persists for an accessible and time-efficient tool capable of accurately distinguishing between PCNSL and other brain lesions, particularly glioma, during surgical procedures.

Deep learning has demonstrated potential in assisting with various aspects of tumor diagnosis, including histological classification[14], molecular typing[15], therapeutic efficacy assessment[16], and prognostic prediction[17] from H&E-stained formalin-fixed paraffin-embedded (FFPE) tissue slides. Similarly, in the context of CNS lesions, deep learning has demonstrated success in classifying, grading, and risk stratification of glioma based on histopathology[18–20]. However, the application of deep learning to frozen samples is still limited and requires further exploration, with most studies focusing on H&E-stained FFPE samples. Histological artifacts present in frozen sections can hinder rapid diagnostic assessments during surgery[21], but a deep learning algorithm may

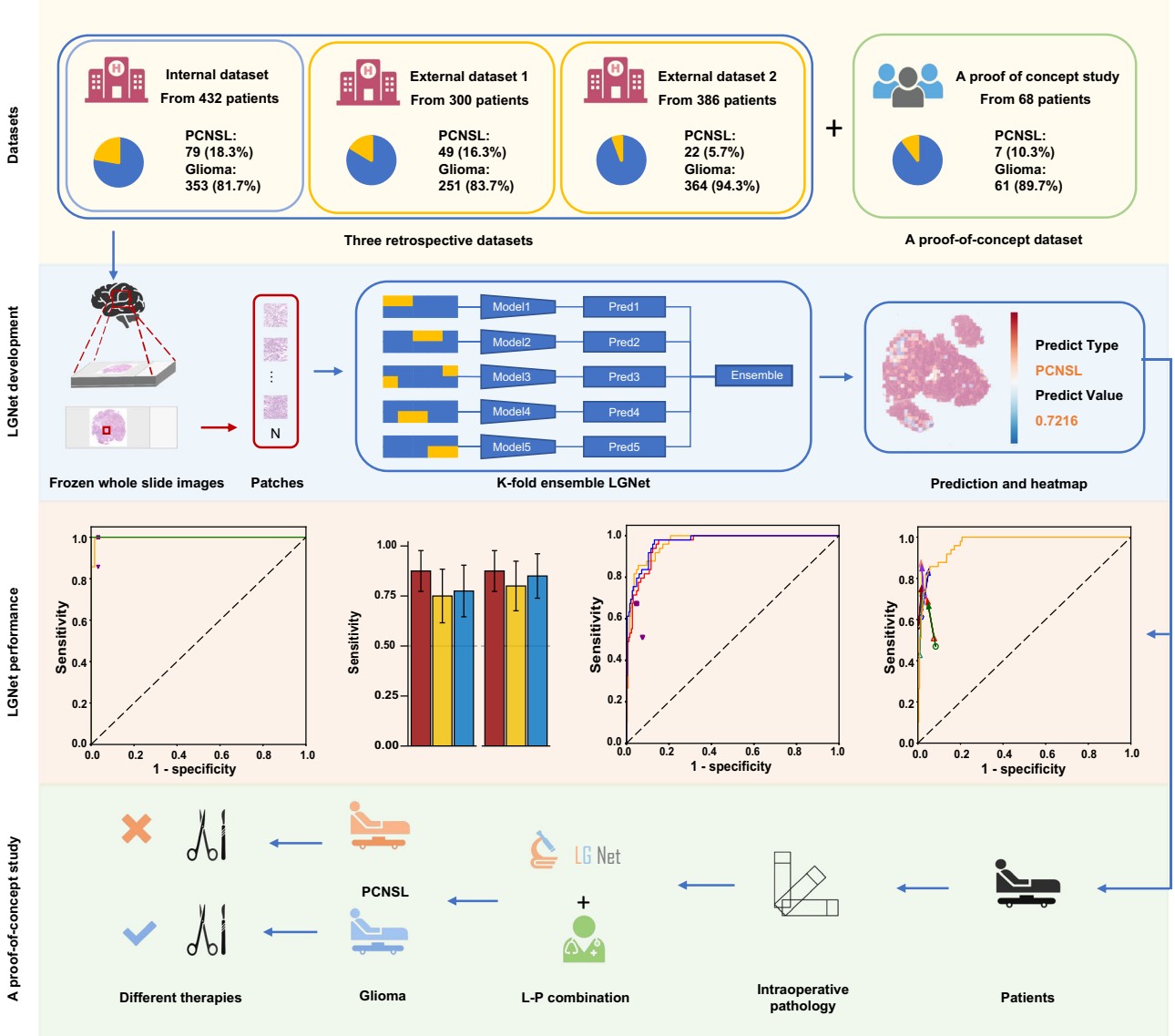

**Fig. 1 | The workflow of the study.** Four retrospective datasets were included in this study. Each whole-slide image was preprocessed and tessellated into non-overlapping tiles of 512 × 512 pixels at 10× magnification. From these tiles, two random tiles of 224 × 224 pixels were extracted and inputted into the LGNet. The LGNet generated tile-level probabilities, which were then averaged to obtain slide-level probabilities for predicting glioma and primary central nervous system lymphoma (PCNSL). The LGNet was constructed by assembling five well-trained

individual classifiers at the output layer. The average probability outputs of these five individual classifiers were used as the prediction of the ensembled model LGNet. The performance of LGNet was developed and validated using an internal dataset and further evaluated on two external retrospective datasets. Finally, the proof-of-concept dataset was used to assess the performance of LGNet in guiding neurosurgeons in decision-making during surgery. Some illustrations were generated with BioRender.com.

improve the quality of whole-slide images (WSIs) from H&E-stained frozen sections, leading to more accurate tumor classification by pathologists[22]. Recent studies have also shown the ability of deep learning models to diagnose thyroid nodules[23] and determine the metastatic status of sentinel lymph nodes in breast cancer[24] from the conventional intraoperative frozen sections, highlighting the potential of frozen samples in developing deep learning models. Thus, we hypothesize a deep learning approach can facilitate the intraoperative diagnosis of brain tumors.

Our study aimed to train and validate a deep learning model capable of accurately differentiating between PCNSL and glioma, including distinguishing PCNSL from other non-PCNSL lesions, using H&E-stained frozen WSIs. We also designed a human-machine fusion approach to improve the diagnostic performance by integrating the abilities of both deep learning models and pathologists. Finally, we conducted a proof-of-concept study to simulate the real-world scenario of frozen diagnosis and assess the practicality of the deep learning model.

## Results

### Study design and patient cohorts

The LGNet, a deep learning model employed for precise prediction of PCNSL and glioma, is detailed in Fig. 1, illustrating the comprehensive workflow. This study encompassed four retrospective cohorts, employing data from both internal and two external cohorts to develop and validate the LGNet model's proficiency in discriminating between PCNSL and non-PCNSL lesions. A thorough description of these datasets is provided in the Supplementary Methods along with additional information in Supplementary Tables 1 and 2.

### Deep learning model performance

To develop a deep learning model capable of predicting PCNSL and glioma from frozen tissue samples, we initially trained ResNet50 using the internal dataset. Subsequently, we assessed this model's performance across both the internal and external cohorts. In the internal cohort, five-fold cross-validation resulted in AUROC value ranging from 0.990 to 1.000, with a mean value of 0.992 at the patient level (Supplementary Table 3). On external cohort 1, LGNet achieved an AUROC of 0.965 (95% CI 0.95–0.98), along with a sensitivity of 0.857 (95% CI 0.73–0.94) and a specificity of 0.912 (95% CI 0.87–0.94) (Table 1 and Fig. 2). On external cohort 2, LGNet obtained an AUROC of 0.972 (95% CI 0.94–1.00), a sensitivity of 0.955 (95% CI 0.77–1.00), and a specificity of 0.868 (95% CI 0.83–0.90) (Table 1 and Fig. 2). Our analysis, detailed in Supplementary Table 4, indicated no significant difference in AUROC with or without color normalization for both external cohorts 1 and 2. In addition, to assess model robustness, we conducted 10 random selections of slides in external cohort 2 based on real-world PCNSL and glioma proportions[25]. The resulting AUROCs were consistent, averaging 0.972 (Supplementary Table 5). When focusing solely on lesions categorized as ideally equivocal on pre-operative imaging, LGNet achieved AUROCs of 0.965 (95% CI 0.93–1.00) and 0.959 (95% CI 0.91–1.01), with sensitivities of 0.875 (95% CI 0.73–0.96) and 0.938 (95% CI 0.70–1.00), and specificities of 0.903 (95% CI 0.81–0.96) and 0.841 (95% CI 0.73–0.92) on external cohorts 1 and 2, respectively (Supplementary Table 6). Moreover, LGNet yielded an AUROC of 0.938 (95% CI 0.88–0.99), a sensitivity of 0.903 (95% CI 0.74–0.98), and a specificity of 0.804 (95%CI 0.66–0.91) in the stereotactic biopsy samples across both external cohorts (Supplementary Table 7). Taken together, our data suggest that LGNet demonstrates comparatively good diagnostic performance, even in scenarios without color normalization or variations in slide selection.

Moreover, for the development and validation of the deep learning model in differentiating PCNSL from non-PCNSL (encompassing glioma and other brain lesions) based on frozen tissue samples, we trained and assessed this model's performance using both internal and external datasets containing PCNSL and non-PCNSL cases. The model exhibited an AUROC ranging from 0.989 to 1.000, with a mean value of 0.996 at the slide level (data not shown). Subsequent evaluation of the external test datasets 1 and 2 revealed promising results. In the former, the model displayed an AUROC reached 0.981 (95% CI 0.97–0.99), along with a sensitivity of 0.980 (95% CI 0.89–1.00) and specificity of 0.847 (95% CI 0.81–0.88). Similarly, in the latter, the model achieved an AUROC of 0.993 (95% CI 0.99–1.00), exhibiting a sensitivity of 1.000 (95% CI 0.85–1.00) and specificity of 0.800 (95% CI 0.76–0.84) (Supplementary Table 8). These results highlight the model's capacity to distinguish PCNSL from non-PCNSL.

### Performance comparison between deep learning model and pathologist

A reader study involving pathologists was conducted on the external datasets to compare the diagnostic performance of the deep learning model with that of eight pathologists, each with varying years of experience in intraoperative neuropathological diagnosis. The pathologists reviewed H&E-stained frozen sections and made diagnoses based on the morphological features. A positive correlation between years of experience and diagnostic performance was observed among the pathologists. In external cohort 1, LGNet had a significantly higher AUROC than pathologists 1, 2, 4, and 5 ($P < 0.001$). Moreover, no statistically significant differences were observed between the AUROC of LGNet and pathologist 3, 6–8, but the value of LGNet's AUROC was higher than that of pathologist 3, 6–8 (Table 1). In external cohort 2, LGNet displayed a significantly superior AUROC compared to pathologists 1 ($P = 0.004$) and 4 ($P < 0.001$). Additionally, although there were no significant differences when comparing the AUROC of LGNet with pathologists 2, 3, 5–8, the AUROC value of LGNet was superior to pathologists 3, 6–8 (Table 1). Across both external cohorts, LGNet consistently demonstrated a higher value of sensitivity than the eight pathologists, especially significantly higher sensitivity compared to pathologists 1, 2, 4, 5, and 7 in external cohort 1 (Table 1). These findings indicate that LGNet' performance significantly surpassed that of pathologists with one year and five years of experience in intraoperative neuropathological diagnosis, and matched or slightly outperform pathologists with a decade years of experience. Focusing solely on cases that were ideally equivocal on preoperative imaging, LGNet exhibited a significantly higher AUROC than pathologists 1, 4 ($P < 0.001$), and 2 ($P = 0.008$) (Supplementary Table 6) in external cohort 1. In external cohort 2, LGNet achieved a significantly better AUROC compared to pathologist 4 ($P < 0.05$) (Supplementary Table 6). When exclusively analyzing cases from the stereotactic biopsy across both external cohorts, LGNet displayed a significantly higher AUROC than pathologists 1 ($P = 0.004$) and 4 ($P < 0.001$) (Supplementary Table 7). Notably, LGNet consistently exhibited higher values of sensitivity than the eight pathologists, in particular, significantly higher sensitivity compared to pathologists 1, 4, 5, and 7 in the external cohort 1 (Supplementary Table 6). In the external test dataset 1 containing PCNSL and non-PCNSL cases, the model showed a significantly higher AUROC than pathologist 3 ($P = 0.012$) (Supplementary Table 8). Collectively, LGNet's outperformance of some pathologists, even those with years of experience, in distinguishing equivocal cases and stereotactic biopsy samples signifies its utility in challenging diagnostic scenarios.

### Performance comparison between LGNet-assisted pathologists and unassisted pathologists

To assess the impact of LGNet on pathologists' diagnostic performance, a re-examination of H&E-stained frozen sections and subsequent re-diagnosis of PCNSL and glioma was conducted post-analysis of LGNet's predictions. Our data showed that LGNet improved the sensitivity of each pathologist in external cohort 1 (Fig. 3). Notably, the diagnostic proficiency of LGNet-assisted pathologists surpassed

**Table 1 | The comparison of LGNet's performance and pathologists' performance on two external cohorts**

| Cohorts | Category | Predictive performance | | | | | | | | |
|---|---|---|---|---|---|---|---|---|---|---|
| | | Sensitivity (95% CI) | P* | Pᵃ | Specificity (95% CI) | P* | Pᵃ | AUROC (95% CI) | P* | Pᵃ |
| External cohort 1 | LGNet | 0.857 (0.73, 0.94) | NA | NA | 0.912 (0.87, 0.94) | NA | NA | 0.965 (0.95, 0.98) | NA | NA |
| | Pathologist 1 | 0.510 (0.36, 0.66) | <0.001 | 0.008 | 0.924 (0.88, 0.95) | 0.73 | 1.00 | 0.840 (0.78, 0.90) | <0.001 | 0.008 |
| | Pathologist 2 | 0.612 (0.46, 0.75) | 0.008 | 0.032 | 0.984 (0.96, 1.00) | <0.001 | 0.008 | 0.843 (0.78, 0.91) | <0.001 | 0.008 |
| | Pathologist 3 | 0.735 (0.59, 0.85) | 0.18 | 0.22 | 0.980 (0.95, 0.99) | 0.001 | 0.008 | 0.922 (0.86, 0.98) | 0.09 | 0.27 |
| | Pathologist 4 | 0.469 (0.33, 0.62) | <0.001 | 0.008 | 0.916 (0.87, 0.95) | 1.00 | 1.00 | 0.754 (0.67, 0.83) | <0.001 | 0.008 |
| | Pathologist 5 | 0.429 (0.29, 0.58) | <0.001 | 0.008 | 0.992 (0.97, 1.00) | <0.001 | 0.008 | 0.900 (0.86, 0.94) | <0.001 | 0.008 |
| | Pathologist 6 | 0.735 (0.59, 0.85) | 0.11 | 0.22 | 0.984 (0.96, 1.00) | <0.001 | 0.008 | 0.952 (0.93, 0.97) | 0.24 | 0.27 |
| | Pathologist 7 | 0.571 (0.42, 0.71) | <0.001 | 0.008 | 0.996 (0.98, 1.00) | <0.001 | 0.008 | 0.948 (0.92, 0.97) | 0.12 | 0.27 |
| | Pathologist 8 | 0.714 (0.57, 0.83) | 0.065 | 0.20 | 0.972 (0.94, 0.99) | 0.003 | 0.009 | 0.923 (0.88, 0.96) | 0.019 | 0.08 |
| External cohort 2 | LGNet | 0.955 (0.77, 1.00) | NA | NA | 0.868 (0.83, 0.90) | NA | NA | 0.972 (0.94, 1.00) | NA | NA |
| | Pathologist 1 | 0.636 (0.41, 0.83) | 0.016 | 0.096 | 0.893 (0.86, 0.92) | 0.37 | 0.74 | 0.876 (0.81, 0.95) | 0.004 | 0.028 |
| | Pathologist 2 | 0.818 (0.60, 0.95) | 0.38 | 1.00 | 0.907 (0.87, 0.93) | 0.12 | 0.48 | 0.896 (0.82, 0.97) | 0.06 | 0.30 |
| | Pathologist 3 | 0.818 (0.60, 0.95) | 0.25 | 1.00 | 0.901 (0.87, 0.93) | 0.20 | 0.60 | 0.948 (0.90, 0.99) | 0.19 | 0.32 |
| | Pathologist 4 | 0.591 (0.36, 0.79) | 0.008 | 0.064 | 0.885 (0.85, 0.92) | 0.56 | 0.74 | 0.812 (0.71, 0.91) | <0.001 | 0.008 |
| | Pathologist 5 | 0.591 (0.36, 0.79) | 0.008 | 0.064 | 0.986 (0.97, 1.00) | <0.001 | 0.008 | 0.875 (0.79, 0.96) | 0.019 | 0.11 |
| | Pathologist 6 | 0.864 (0.84, 0.99) | 0.50 | 1.00 | 0.940 (0.91, 0.96) | <0.001 | 0.008 | 0.917 (0.84, 0.99) | 0.10 | 0.32 |
| | Pathologist 7 | 0.773 (0.55, 0.92) | 0.13 | 0.65 | 0.967 (0.94, 0.98) | <0.001 | 0.008 | 0.917 (0.84, 0.99) | 0.11 | 0.32 |
| | Pathologist 8 | 0.864 (0.65, 0.97) | 0.50 | 1.00 | 0.915 (0.88, 0.94) | 0.036 | 0.18 | 0.953 (0.91, 0.99) | 0.08 | 0.32 |

The difference comparison between AUROCs was used in Delong's test. The McNemar test was used to compare the statistical differences in sensitivity and specificity. The sample size to derive statistics is n = 300 (external cohort 1) and n = 386 (external cohort 2) independent patient samples for each variable. P value is two-sided. Pᵃ adjusted P value with the FDR method. The data have been provided in the Source Data file.
95% CI 95% confidence intervals, AUROC the area under the receiver operating characteristic, NA not applicable.
*indicates the comparison of the difference between each pathologist and the LGNet.

a

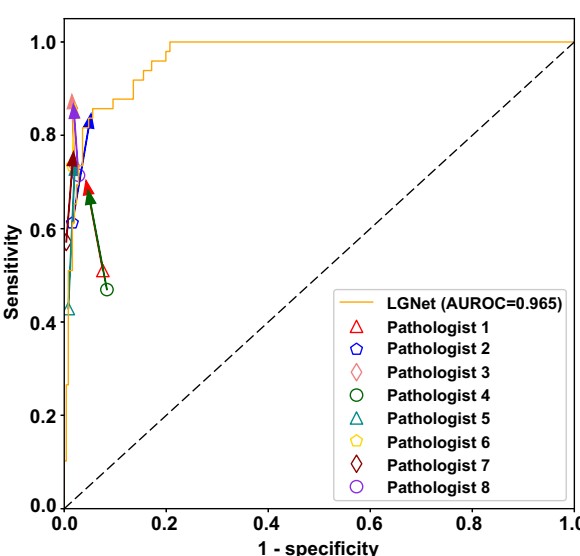

b

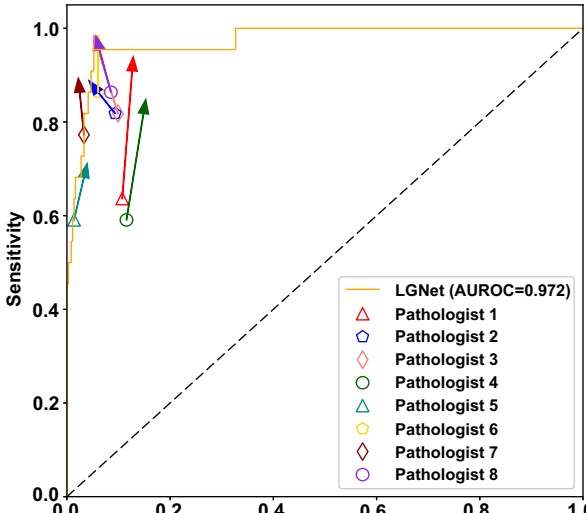

**Fig. 2 | The diagnostic performance of the LGNet and pathologists on two external datasets.** External cohort 1 (**a**); external cohort 2 (**b**). LGNet, a deep learning model designed for distinguishing primary central nervous system lymphoma (PCNSL) from glioma; Pathologists 1 and 4, each with one year of experience in intraoperative neuropathological diagnosis; Pathologists 2 and 5, each possessing five years of experience in intraoperative neuropathological diagnosis;

Pathologists 3, 6, 7, and 8, each exhibiting expertise up to ten years in intraoperative neuropathological diagnosis; The points at the base of each arrow represent the performance of each pathologist unassisted by LGNet, while the arrow indicates the change in their performance with LGNet's assistance; AUROC the area under the receiver operating characteristic curve.

that of pathologists operating without LGNet support. In external cohort 1, the AUROCs of eight LGNet-assisted pathologists ranged between 0.900 (95% CI: 0.85–0.95) and 0.961 (95% CI: 0.94–0.99). These LGNet-assisted pathologists displayed significantly higher AUROCs than their unassisted counterparts ($P < 0.05$) (Fig. 3). Similarly, in external cohort 2, eight LGNet-assisted pathologists demonstrated AUROCs ranging from 0.952 (95% CI: 0.92–0.98) to 0.978 (95% CI: 0.95–1.00), significantly surpassing the performance of pathologists 1, 4, 5, and 8 working independently (Supplementary Fig. 1). Focusing exclusively on cases with equivocal imaging diagnoses or from stereotactic biopsy within both external cohorts, LGNet-assisted pathologists also outperformed pathologists operating without LGNet assistance (Supplementary Tables 9–11). Remarkably, within external cohort 1, LGNet-assisted pathologists 1 and 4, labeled as less experienced due to their one year of intraoperative neuropathological expertise, exhibited comparable AUROCs to those of pathologists 2 and 5, characterized as experienced professionals with five years of experience in diagnosing neuropathological frozen slides (Fig. 3). Similarly, in external cohort 2, the AUROCs of LGNet-assisted pathologists 1 and 4 were comparable to not only pathologists 2 and 3 but also pathologists 5–8, possessing a decade of experience (Supplementary Fig. 1). Consistent trends were observed when analyzing solely cases with equivocal imaging diagnosis or from the stereotactic biopsy (Supplementary Tables 9–11). Thus, LGNet exhibits the potential to provide expert-level assistance intraoperative diagnoses.

**Human-machine fusion**
To evaluate the effectiveness of deep learning model in intraoperative diagnosis, we examined the human-machine fusion on two external datasets. We presented both the original and modified fusion workflows in Supplementary Fig. 2. The original fusion of LGNet and pathologists 1–8 on external cohort 1 yielded AUROCs ranging from 0.942 (95% CI: 0.91–0.97) to 0.983 (95% CI: 0.97–0.99), which were comparable to or marginally higher than LGNet's individual performance (0.965) (Figs. 4 and 5). In contrast, the modified fusion, involving LGNet and pathologists 1–8, demonstrated AUROCs ranging from

0.965 (95% CI: 0.94–0.99) to 0.986 (95% CI: 0.98–1.00), equaling or outperforming LGNet (0.965). Notably, the modified fusion predictions made by pathologists 6–8 in conjunction with LGNet significantly outperformed LGNet alone, and the performance of pathologist 8 in the modified fusion approach distinctly surpassed that of the original fusion (Figs. 4 and 5). On external cohort 2, the modified fusion predictions made by pathologists 3, and 5 through 8, in collaboration with LGNet, significantly outperformed LGNet alone. The AUROC value of the modified fusion prediction of pathologist 1 and LGNet is higher than LGNet, in spite of no significant difference (Fig. 4 and Supplementary Fig. 3). Importantly, in external cohort 1, the modified fusion approach achieved sensitivities ranging from 0.755 to 0.878 and specificities ranging between 0.948 and 0.976 (Fig. 5). In external cohort 2, the corresponding sensitivity ranged from 0.909 to 0.955. Specificity between 0.901 and 0.964 (Supplementary Fig. 3). The specificity of the original and modified fusion is significantly higher than the model's on both two external cohorts ($P < 0.05$). Specifically considering cases with equivocal imaging diagnosis in external cohorts 1 and 2, the modified fusion's AUROC matched or outperformed the original fusion's (Supplementary Figs. 4 and 5). In external cohort 1, the modified fusion predictions made by pathologists 6 and 8, in collaboration with LGNet, exceeded LGNet alone, and the performance of pathologist 8, in conjunction with LGNet, in the modified fusion approach was superior to that of the original fusion. Notably, the performance of pathologist 4 in the modified fusion approach distinctly surpassed that of the original fusion and the specificity of modified fusion of pathologist 3 or 8, in combination with LGNet, is distinctly higher than LGNet alone (Supplementary Fig. 4). Similarly, on external cohort 2, the performance of pathologist 4, in conjunction with LGNet, in the modified fusion approach was superior to that of the original fusion, and the specificity of modified fusion combining pathologist 5 or 7 with LGNet significantly exceeded LGNet alone (Supplementary Fig. 5). When specifically analyzing cases from stereotactic biopsy in both external cohorts, the modified fusion predictions made by pathologist 3 or 6, in collaboration with LGNet, outperformed LGNet alone, and modified fusion predictions made by pathologist 6, in combination

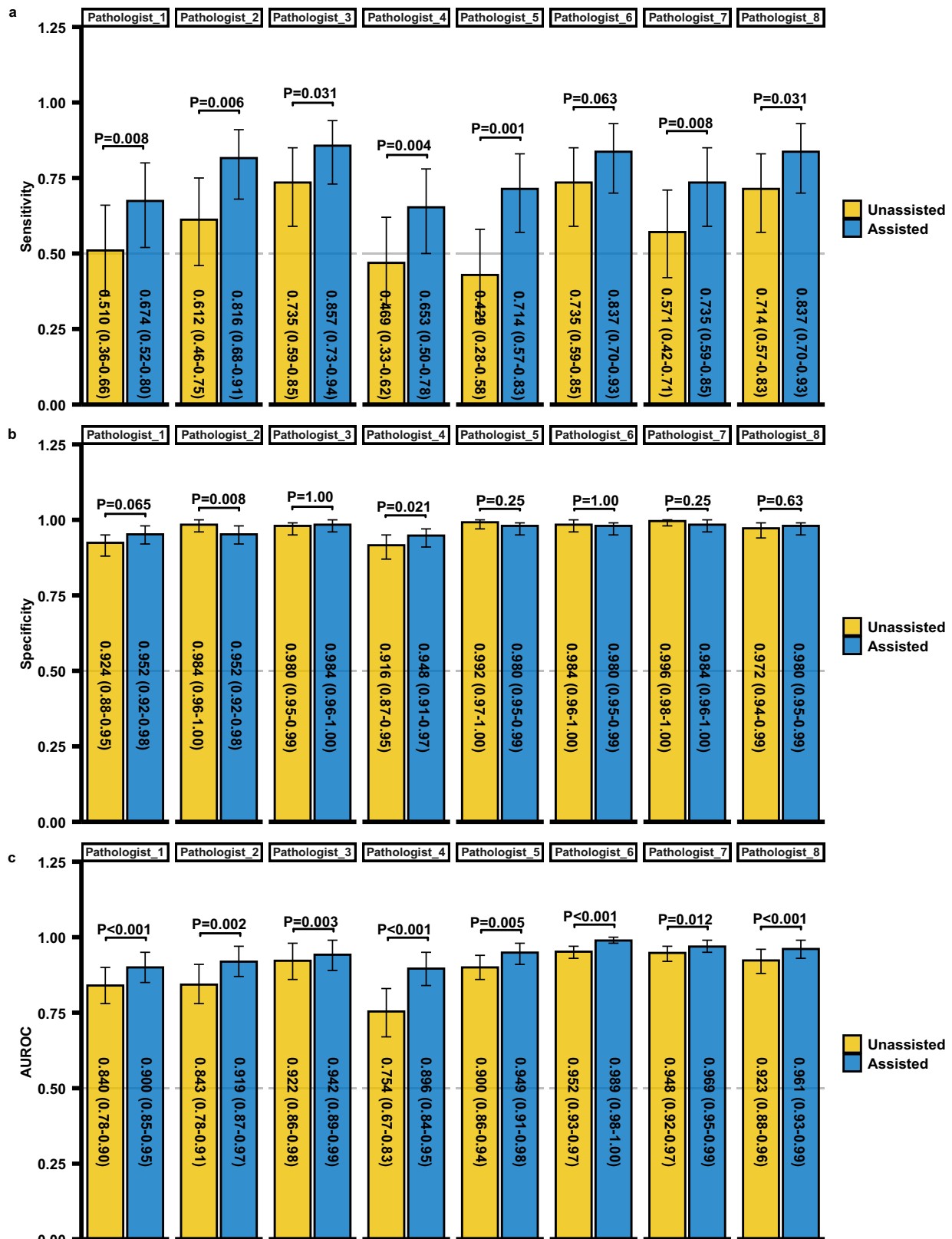

with LGNet, distinctly surpassed LGNet alone, and the performance of pathologist 4, in conjunction with LGNet, in the modified fusion approach was distinctly higher than that of the original fusion. Especially, the specificity of modified fusion combining pathologist 1 or 3 with LGNet significantly exceeded LGNet alone (Supplementary Fig. 6). Overall, our data indicate that the modified human-machine fusion

approach holds promise for enhancing intraoperative differentiation between PCNSL and glioma, potentially improving diagnostic performance to some extent.

Additionally, concerning the external test datasets, the original fusion with pathologist 3 yielded an AUROC of 0.988 (95% CI: 0.98–1.00; $P = 0.25$), while the modified fusion reached an AUROC of

**Fig. 3 | The diagnostic performance of each pathologist was compared with and without the assistance of LGNet on the external cohort 1.** The performance was measured using three parameters: Sensitivity (**a**), Specificity (**b**), and AUROC (**c**). Pathologists 1 and 4, with one year of experience in intraoperative neuropathological diagnosis; Pathologists 2 and 5, having five years of experience in intraoperative neuropathological diagnosis; Pathologists 3, 6, 7, and 8, having up to ten years of experience in neuropathological intraoperative diagnosis; Pathologist (unassisted), working without the aid of LGNet; Pathologist (assisted), assisted by LGNet; AUROC, the area under the receiver operating characteristic; The error bars are the 95% CI, with the measure of the histogram being the sensitivity, specificity and AUROC of each variable. The sample size to derive statistics is $n = 300$ independent patient samples for each variable. The difference comparison between AUROCs was used in Delong's test. The McNemar test was used to compare the statistical differences in sensitivity and specificity. The $P$ value was evaluated from a two-sided test. Adjustments were made for multiple comparisons. The data have been provided in the Source Data file.

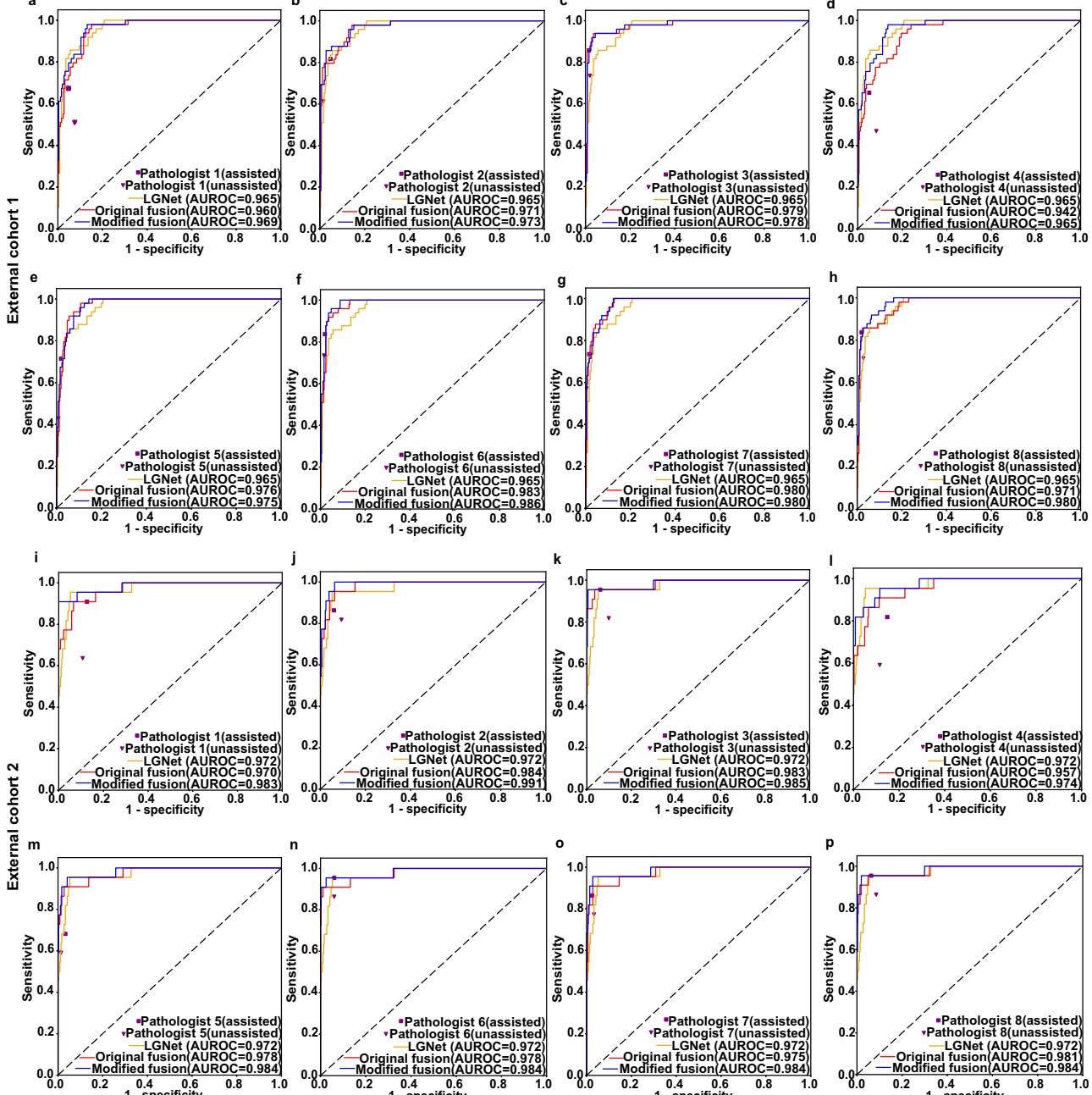

**Fig. 4 | Fusion of the LGNet and pathologists on two external datasets.** External cohort 1 (**a**–**h**); external cohort 2 (**i**–**p**). Pathologists 1 and 4, with one year of experience in intraoperative neuropathological diagnosis; Pathologists 2 and 5, having five years of experience in intraoperative neuropathological diagnosis; Pathologists 3, 6, 7, and 8, having up to ten years of experience in intraoperative neuropathological diagnosis; Pathologist (unassisted), working without the aid of LGNet; Pathologist (assisted), assisted by LGNet; AUROC, the area under the receiver operating characteristic; Original fusion, the fusion from LGNet's prediction and pathologist's original diagnosis (pathologist's diagnosis without the aid of LGNet); Modified fusion, the fusion from LGNet's prediction and pathologist's modified diagnosis (pathologist's diagnosis with the aid of LGNet).

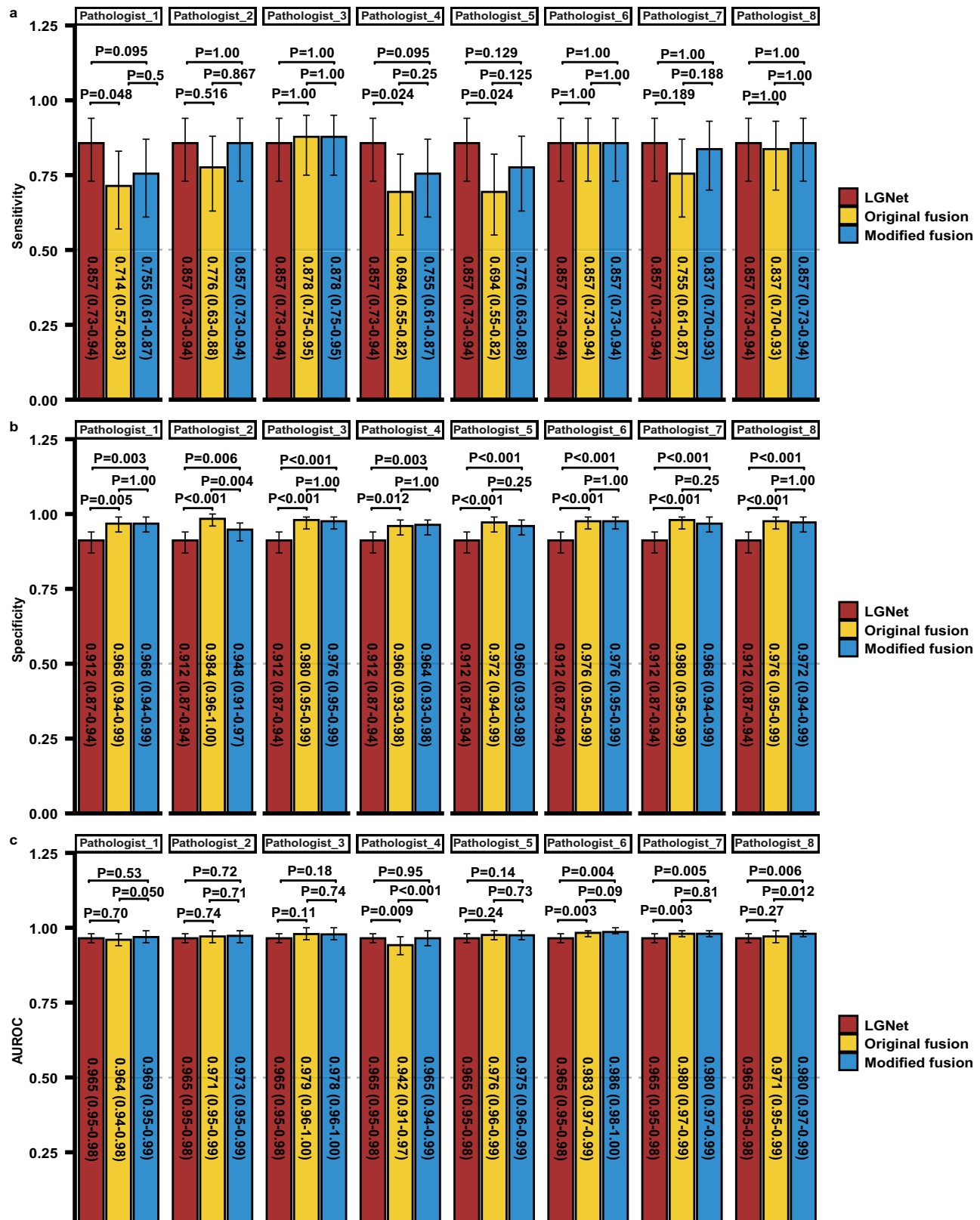

0.989 (95% CI: 0.98–1.00; $P = 0.16$). Although no statistically significant differences were found, the performance of the fusion approach remains slightly higher than that of the model in distinguishing PCNSL from non-PCNSL (0.981). Comparable outcomes were observed in the external test dataset 2. Additionally, no statistically significant differences were shown between the sensitivity of the fusion and the model.

However, the specificity of the fusion approach is significantly higher than that of the model ($P < 0.001$) on two external test cohorts. (Supplementary Table 8). These results suggest that the human-machine fusion approach may also be suitable for distinguishing PCNSL from non-PCNSL due to a significant increase in specificity during surgical procedures.

**Fig. 5 | The comparison of LGNet fusion prediction and pathologist prediction on the external cohort 1.** The performance was measured using three parameters: Sensitivity (**a**), Specificity (**b**), and AUROC (**c**). Pathologists 1 and 4, with one year of experience in intraoperative neuropathological diagnosis; Pathologists 2 and 5, having five years of experience in intraoperative neuropathological diagnosis; Pathologists 3, 6, 7, and 8, having up to ten years of experience in intraoperative neuropathological diagnosis; Pathologist (unassisted), working without the aid of LGNet; Pathologist (assisted), assisted by LGNet; AUROC, the area under the receiver operating characteristic; Original fusion, the fusion from LGNet's prediction and pathologist's original diagnosis (pathologist's diagnosis without the aid of LGNet); Modified fusion, the fusion from LGNet's prediction and pathologist's modified diagnosis (pathologist's diagnosis with the aid of LGNet). The error bars are the 95%CI, with the measure of the histogram being the sensitivity, specificity, and AUROC of each variable. The sample size to derive statistics is *n* = 300 independent patient samples for each variable. The difference comparison between AUROCs was used in Delong's test. The McNemar test was used to compare the statistical differences in sensitivity and specificity. The *P* value was evaluated from a two-sided test. Adjustments were made for multiple comparisons. The data have been provided in the Source Data file.

**Table 2 | The association between LGNet's prediction and the morphological features on external cohorts by logistic regression models**

| Features | External cohort 1 | | External cohort 2 | |
|---|---|---|---|---|
| | Univariate | | Univariate | |
| | OR (95% CI) | *P* | OR (95% CI) | *P* |
| Perivascular cuffing of tumor cells | 16.71 (3.45, 80.88) | <0.001 | 7.88E9 (0.00, +∞) | 1.00 |
| Monomorphic nuclei | 22.60 (10.15, 50.34) | <0.001 | 6.71 (3.27, 13.77) | <0.001 |
| Prominent nucleoli | 27.97 (10.82, 72.35) | <0.001 | 11·66 (4.75, 28.60) | <0.001 |
| Scant cytoplasm | 11.80 (6.23, 22.36) | <0.001 | 2.70 (1.54, 4.75) | 0.001 |
| Poorly cohesive | 12.24 (6.47, 23.17) | <0.001 | 12.63 (5.18, 30.76) | <0.001 |
| Apoptosis | 6.05E9 (0.00,+∞) | 1.00 | 8.13E9 (0.00, +∞) | 1.00 |
| Fibrillary background | 0.22 (0.12, 0.40) | <0.001 | 0.40 (0.22, 0.75) | 0.004 |
| Variation in nuclear shape and size with accompanying hyperchromasia | 0.06 (0.03, 0.13) | <0.001 | 0.18 (0.09, 0.37) | <0.001 |
| Microvascular proliferation | 0.19 (0.05, 0.82) | 0.026 | 0.43 (0.22, 0.84) | 0.013 |
| Necrosis | 0.35 (0.08, 1.53) | 0.16 | 0.64 (0.26, 1.58) | 0.33 |

The association between LGNet prediction and morphological features was analyzed by logistic regression models. *P* value is two-sided. The sample size to derive statistics is *n* = 300 (external cohort 1) and *n* = 386 (external cohort 2) independent patient samples for each variable. The data have been provided in the Source Data file.
*95% CI* 95% confidence intervals, *OR* odds ratio.

## Association between histological characteristics and LGNet prediction of PCNSL

To delve deeper into the mechanisms of the deep learning model, we conducted a logistic regression analysis to investigate the correlation between LGNet's predictions and histopathological features. Our univariate analysis demonstrated significant associations between LGNet's prediction of PCNSL and several histopathological characteristics, including monomorphic nuclei, prominent nucleoli, scant cytoplasm, and poorly cohesive, across both external cohorts (All *P* < 0.001, Table 2). Furthermore, we employed prediction heatmaps generated by LGNet to delineate regions within the slides that received high or low prediction scores for PCNSL and glioma. Notably, regions exhibiting histopathological attributes such as monomorphic nuclei, prominent nucleoli, scant cytoplasm, poor cohesiveness, and perivascular cuffing of tumor cells tended to obtain high scores for PCNSL. Conversely, sections depicting features like a fibrillary background, variations in nuclear shape and size with hyperchromasia, and microvascular proliferation were more closely associated with lower scores for glioma (Fig. 6). These findings provide valuable insights into the underlying mechanisms of LGNet's predictions, highlighting the histopathological features pivotal in accurately identifying PCNSL cases.

## Misdiagnosis from LGNet

To gain a more comprehensive understanding of the deep learning model's performance, we analyzed the cases where LGNet misclassified instances of PCNSL and glioma. On two external cohorts, 54 out of 78 slides misdiagnosed by LGNet were correctly diagnosed by all pathologists. In external cohort 1, LGNet exhibited misdiagnoses in a total of 29 slides, encompassing 7 PCNSLs and 22 gliomas. In external cohort 2, LGNet misdiagnosed 49 slides, comprising 1 PCNSL and 48 gliomas (Supplementary Fig. 7). Among the 22 cases misdiagnosed as PCNSL in external cohort 1 by LGNet, 4 (18.2%) and 3 (13.6%) displayed PCNSL-like features, including monomorphic nuclei and prominent nucleoli, respectively. In addition, among the 48 instances misclassified as PCNSL in external cohort 2, 2 (4.2%) cases exhibited the feature of monomorphic nuclei (Supplementary Fig. 8). Supplementary Table 12 provided additional evidence, showing a significant association between histomorphological attributes of glioma, misdiagnosed as PCNSL and PCNSL-like features, such as monomorphic nuclei and prominent nucleoli (*P* < 0.001). These observations shed light on the specific characteristics contributing to LGNet misdiagnosis of PCNSL.

## Performance on the proof-of-concept study

To gauge the practical use of LGNet in clinical settings, we conducted a proof-of-concept study at our facility. The process of predicting PCNSL and glioma using the deep learning model through the online pathological decision platform was elucidated in Supplementary Movie 1. LGNet exhibited a significantly higher AUROC than pathologist A (0.998 vs. 0.821, *P* = 0.005). No statistically significant differences were observed between the AUROC of LGNet and pathologist B (0.998 vs. 0.972, *P* = 0.251) (Table 3 and Fig. 7). When aided by LGNet, pathologist A achieved an increased AUROC of 0.991 (95% CI: 0.98–1.01) compared to working alone (*P* = 0.003), while pathologist B obtained an elevated AUROC of 0.991 (95% CI: 0.97–1.01), with no significant differences between LGNet-assisted and unassisted AUROCs (*P* = 0.26) (Supplementary Table 13 and Fig. 7). Although no statistical differences were found between the performance of the combination of LGNet and pathologist (L-P) prediction, or human-machine fusion and LGNet alone (1.000 vs. 0.998, *P* = 0.48; 1.000 vs. 0.998, *P* = 0.48) (Table 3 and Fig. 7), these findings suggest that LGNet can assist pathologist with less years of experience in improving diagnostic

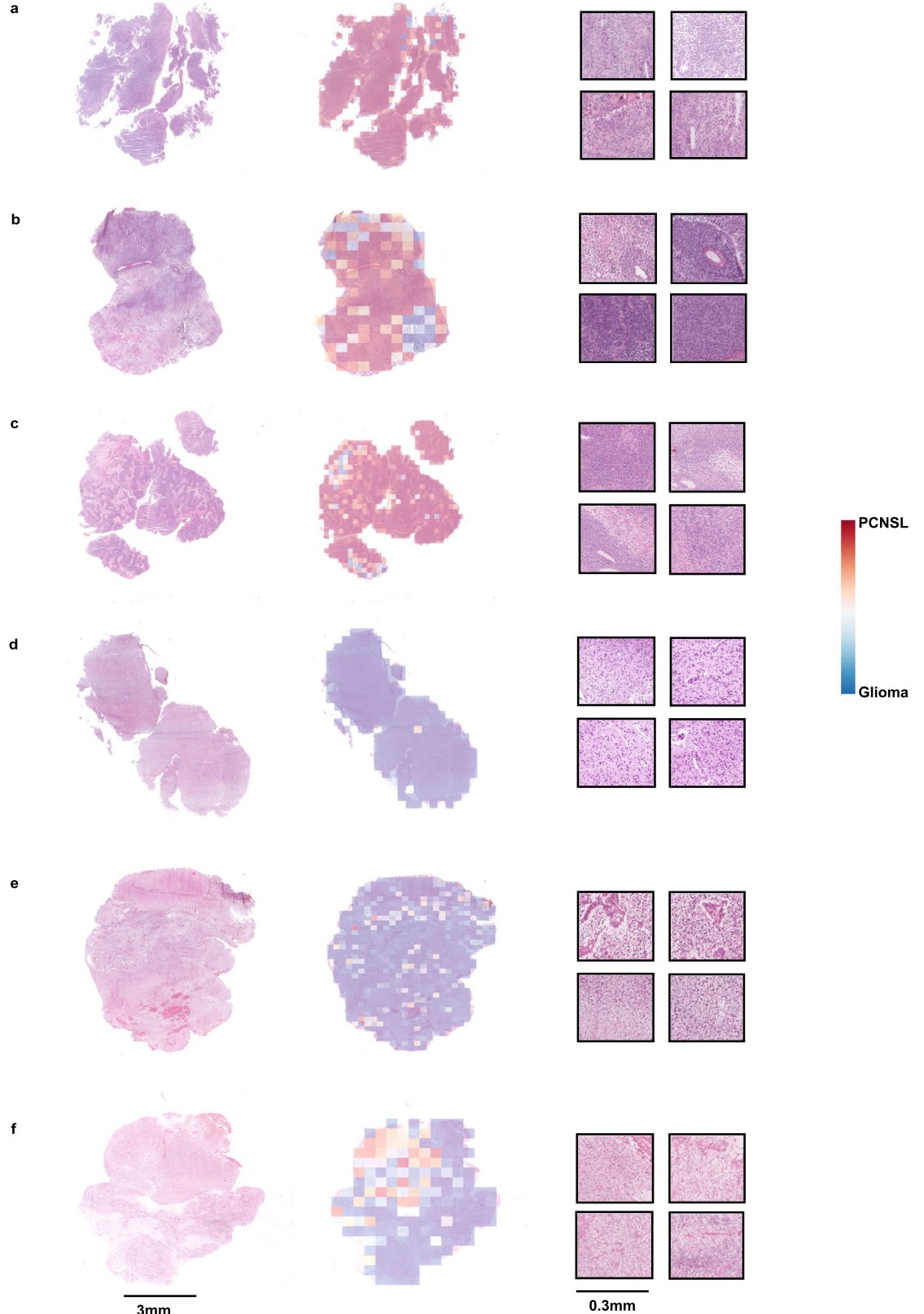

**Fig. 6 | The LGNet model successfully predicted cases of lymphoma and glioma in internal cohort, external cohort 1, and external cohort 2.** The histological images of the patients with primary central nervous system lymphoma (PCNSL) (**a**–**c**) and glioma (**d**–**f**) are shown in the left column. The heatmaps overlapped on the whole-slide images (WSIs) in the middle column indicated the tissue tiles that LGNet predicted as PCNSL with a high score (reddish color) or as glioma with a low score (bluish color). The tiles with a high score for PCNSL were primarily localized in areas of perivascular cuffing of tumor cells, monomorphic nuclei, prominent nucleoli, scant cytoplasm, and poor cohesiveness (tiles at 10× magnification in the right column). Similarly, the tiles with a low score for glioma were more likely to be found in areas of fibrillary background, nuclear shape and size variation with hyperchromasia, and microvascular proliferation (tiles at 10× magnification in the right column). All results were reproducible and consistent, demonstrating the reliability and stability of the LGNet model.

**Table 3 | The comparison of the performance of LGNet, pathologist, and LGNet-pathologist combination on the proof-of-concept cohort**

| Category | Diagnostic metrics | | | | | | | | |
|---|---|---|---|---|---|---|---|---|---|
| | Sensitivity (95% CI) | P* | Pᵃ | Specificity (95% CI) | P* | Pᵃ | AUROC (95% CI) | P* | Pᵃ |
| LGNet | 0.857(0.42,1.00) | NA | NA | 0.984 (0.91,1.00) | NA | NA | 0.998 (0.99,1.01) | NA | NA |
| Pathologist A | 0.571 (0.18,0.90) | 0.63 | 1.00 | 0.836 (0.72,0.92) | 0.012 | 0.036 | 0.821 (0.70,0.94) | 0.005 | 0.020 |
| Pathologist B | 0.857 (0.42,1.00) | 1.00 | 1.00 | 0.967 (0.89,1.00) | 1.00 | 1.00 | 0.972 (0.92,1.02) | 0.25 | 0.75 |
| L-PA Combination | 1.000 (0.59,1.00) | NA | NA | 1.000 (0.94,1.00) | NA | NA | 1.000 (1.00,1.00) | 0.48 | 0.96 |
| L-PB Combination | 1.000 (0.59,1.00) | NA | NA | 0.984 (0.91,1.00) | 1.00 | 1.00 | 1.000 (1.00,1.00) | 0.48 | 0.96 |

L-P combination, the combination of LGNet and Pathologist with the assistance of LGNet; Pathologist A, the pathologist with one year of experience in intraoperative diagnosis; Pathologist B, the pathologist with up to 10 years of experience in intraoperative diagnosis; L-PA combination, the combination of LGNet and Pathologist A with the assistance of LGNet; L-PB combination, the combination of LGNet and Pathologist B with the assistance of LGNet; The difference comparison between AUROCs was used to Delong's test. The McNemar test was used to compare the statistical differences in sensitivity and specificity. The sample size to derive statistics is *n* = 68 independent patient samples for each variable. *P* value is two-sided. *Pᵃ* adjusted *P* value with FDR methodThe data have been provided in the Source Data file.
*95% CI* 95% confidence intervals, *AUROC* the area under the receiver operating characteristic, *NA* not applicable.
*indicates the comparison of the difference between the LGNet and the other categories (pathologists and L-P combination).

a

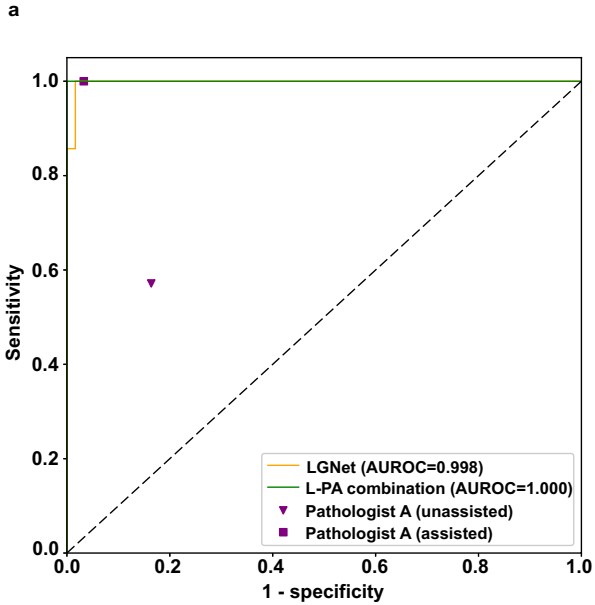
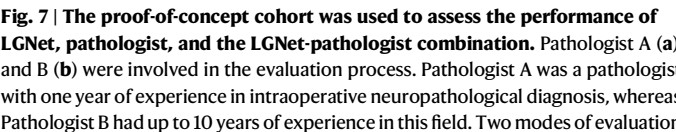

b

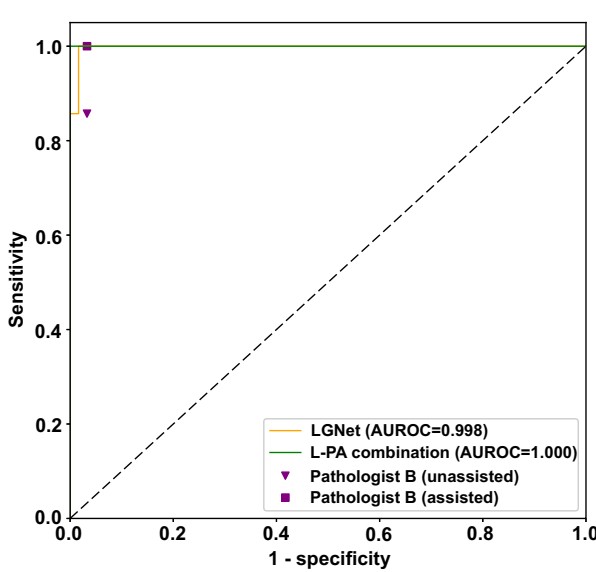

**Fig. 7 | The proof-of-concept cohort was used to assess the performance of LGNet, pathologist, and the LGNet-pathologist combination.** Pathologist A (**a**) and B (**b**) were involved in the evaluation process. Pathologist A was a pathologist with one year of experience in intraoperative neuropathological diagnosis, whereas Pathologist B had up to 10 years of experience in this field. Two modes of evaluation were used: Pathologist (unassisted) and Pathologist (assisted), with the latter being aided by LGNet. The performance was measured using AUROC (area under the receiver operating characteristic). The LGNet-Pathologist A combination (L-PA) and LGNet-Pathologist B combination (L-PB) were also evaluated.

performance. Moreover, regardless of diagnostic experience, the L-P combination fusion remains a viable alternative to LGNet's diagnosis.

## Discussion

In this study, we devised a deep learning approach to aid the intraoperative diagnosis of brain tumors using conventional frozen H&E slides. Our generated deep learning model, LGNet, presents substantial advantages in addressing challenges experienced by pathologists when differentiating PCNSL from glioma, alongside other lesions, during surgical procedures. LGNet effectively predicts PCNSL directly from conventional H&E-stained frozen WSIs, showcasing enhanced performance validated across two external cohorts. Our study unveils a promising approach for intraoperative brain tumor diagnosis through AI technology.

Accurate identification of PCNSL in brain tumor patients holds the potential to prevent unnecessary surgical resection, thus minimizing the risk of brain tissue damage and improving patient care. Our model displayed robust predictive capabilities in accurately distinguishing PCNSL from glioma and other CNS lesions, especially in cases with ambiguous preoperative imaging and predominantly through stereotactic biopsies, presenting notable challenges during intraoperative diagnosis. Surprisingly, LGNet-assisted pathologists displayed significantly higher diagnostic performance than their unassisted counterparts. Furthermore, several pathologists, assisted by LGNet even with just one year of experience, demonstrated comparable or slightly superior performance compared to their unassisted counterparts with five years of experience. Moreover, their performance could rival that of pathologists with ten years of experience to some extent. This assistance is particularly evident in lesions classified as ideally equivocal on preoperative imaging or obtained via stereotactic biopsy, suggesting that LGNet significantly enhances several pathologists' performance during surgery. Thus, this approach holds promise for delivering expert-level intraoperative diagnosis, particularly in settings with limited neuropathology resources, while also enhancing diagnostic performance in well-equipped centers. Interestingly, analysis of LGNet's misdiagnosed slides revealed that 54 out of 78 slides were

correctly diagnosed by all pathologists. Notably, 7 out of 49 PCNSL cases were entirely corrected by different pathologists with five or ten years of experience in intraoperative neuropathological diagnosis in external cohort 1. Aligning with previous studies[26], this underscores complementary strengths between pathologists and the model in intraoperative diagnosis of PCNSL and the potential for improving diagnostic performance by integrating deep learning algorithms with pathologists' expertise.

To explore the deep learning mode's efficacy in time-sensitive intraoperative diagnosis, we employed a human-machine fusion strategy, combining a deep learning model and a pathologist based on their prediction uncertainties, as outlined in our previous study[27]. Our findings indicate that the modified fusion approach could improve the performance for several inexperienced and experienced pathologists collaborating with LGNet in differentiating PCNSL from glioma during surgery, even in complex scenarios for experienced pathologists. Significantly, this fusion strategy could be employed to distinguish PCNSL from non-PCNSL, resulting in a noteworthy increase in specificity. Thus, the human-machine fusion approach could emerge as a promising avenue, enhancing the capabilities of several experienced pathologists and assisting inexperienced ones during challenging cases.

Moreover, We devised an online pathological decision support platform to evaluate LGNet's real-world applicability. Our study demonstrated LGNet's robust performance in identifying PCNSL and assisting pathologists in accurately improving diagnostic performance. Furthermore, the LGNet-Pathologist combination, or human-machine fusion, displayed potential in intraoperative diagnosis and may be used as a viable alternative to LGNet's diagnosis. More importantly, we found that LGNet's performance without color normalization remained comparable to that with color normalization, saving time while preserving the model's favorable diagnostic performance. Overall, our study suggests that our model is suitable for intraoperative diagnosis within strict time constraints.

Deep learning models are often considered "black boxes" due to the lack of transparency in their decision-making process[28,29]. To shed light on LGNet's interpretation, we conducted logistic regression analysis to investigate the association between well-known morphological characteristics and LGNet's predictions. While all ten characteristics investigated have established connections with differentiating PCNSL from glioma, only three characteristics (presence of monomorphic nuclei, prominent nucleoli, and scant cytoplasm) were significantly linked to LGNet's prediction for PCNSL through multivariable analysis. This indirect insight into LGNet's interpretability enhances pathologists' confidence in the model's predictions. Our findings could contribute to improving the transparency of deep neural networks before their integration into routine clinical workflows.

However, our study has some limitations that merit attention[30,31]. Firstly, both the proof-of-concept and internal cohorts were drawn from the same medical center, potentially introducing bias in the proof-of-concept study. Therefore, a prospective study encompassing multiple medical centers is warranted to further evaluate LGNet's real-world potential. Secondly, refinement of the online platform is necessary before deployment for clinical use. Thirdly, we did not use these modern tools, including radiomics[12], genetic biomarkers[32], and cell-free DNA[33] to select doubtful cases for analysis. Fourthly, this is only a proof-of-concept study that needs for future analyses on a suitable number of pathologists and on better-selected cases (small biopsies and difficult neuroimaging interpretation) to draw more reliable conclusions. Lastly, while the logistic regression model provides biological interpretability for the deep learning model, more advanced visual methods, such as high-resolution class activation mapping[34], could offer further insights into the model functioning.

In summary, our study introduces the LGNet deep learning model for differentiating PCNSL from glioma on H&E-stained frozen WSIs and demonstrated its superior performance across two external cohorts. LGNet significantly outperforms some board-certified pathologists and supports pathologists, irrespective of their experience, in improving diagnostic accuracy. Notably, the model also improves diagnostic accuracy in discriminating PCNSL from non-PCNSL, especially glioma. Importantly, the human-machine fusion approach, particularly the modified fusion, further could enhance overall diagnostic performance. These findings, validated with the proof-of-concept cohort, provide guidance for neurosurgeons in informed decision-making for managing patients with different malignant brain tumors during surgery.

## Methods
### Study participants
All patient-related information obtained was ethically approved by the Institutional Ethics Committee at Sun Yat-sen University Cancer Center, and the reference number of the committee is SL-B2022-613-01. The informed consent was waived because patients were not directly recruited for this study. The gender of participants was considered in the study design and reported in the characteristics of different cohorts, and determined on self-report. A specific gender-based analysis a priori was not performed.

To develop and validate the LGNet model, we conducted a retrospective study using three independent cohorts comprising frozen section images acquired between January 1, 2014, and August 31, 2021. These cohorts included an internal cohort from Sun Yat-sen University Cancer Center, alongside two external cohorts: one from Zhujiang Hospital (external cohort 1) and the other from Nanfang Hospital of Southern Medical University and The First Affiliated Hospital of Sun Yat-sen University (external cohort 2). Subsequently, to assess the practicality of the LGNet in clinical settings, we recruited a proof-of-concept cohort from September 1, 2021, to March 1, 2022, at Sun Yat-sen University Cancer Center. In addition, to further expand and validate the deep learning model's capacity to differentiate PCNSL from non-PCNSL (including glioma and other brain lesions), we broadened our dataset to encompass frozen section images of various brain lesions like medulloblastoma, central neurocytoma, metastatic cancer, and inflammation lesions (see the supplementary methods for details). Further details regarding dataset expansion and inclusion/exclusion criteria for patients are outlined in the supplementary methods.

### Slides scanning and WSIs preprocessing
In both the internal and external datasets, we gathered representative H&E-stained frozen slides per patient, featuring specimen sizes ranging from $0.2\,cm \times 0.2\,cm \times 0.1\,cm$ to $1.5\,cm \times 1.5\,cm \times 0.3\,cm$. Employing the Aperio AT2 scanner (Leica Biosystems; Wetzlar, Germany) at 40× magnification (0.25 μm/pixel), WSIs, were acquired and subsequently stored in SVS format. These WSIs were then divided into non-overlapping $512 \times 512$ pixel windows using the openslide library. To prepare inputs for the model, two tiles, each sized at $224 \times 224$ pixels, were randomly selected from each window. Further specifics on the processing of WSIs are delineated in the Supplementary methods.

### Deep learning model development
Following image preprocessing, we initiated training of the ensemble binary deep learning model classifier using the internal dataset to accurately distinguish between PCNSL or glioma, and PCNSL or non-PCNSL. The cohort underwent a random division into three distinct sets: a training set, a validation set, and an internal test set. Importantly, there was a complete absence of overlap among patients or slides across these sets. Employing a five-fold cross-validation scheme, we derived either slide-level or patient-level probability via the deep learning model. This probability was subsequently dichotomized to

achieve the definitive binary classification of patients as either PCNSL or glioma and either PCNSL or non-PCNSL. For further details on model development, refer to the Supplementary methods.

## Deep learning model evaluation

To assess the efficacy of our deep learning model, we performed both internal and external assessments on the respective datasets. Internally, the internal cohort underwent division into five folds. Four folds were utilized to train an ensemble classifier, while the remaining fold was dedicated to serve as the internal test set in each iteration. This procedure involved further subdivision of the data into five new folds, with each fold employed for training an individual classifier, as described previously. The ensemble classifier thus constructed was then evaluated at both slide and patient levels using the internal test set. This process was repeated five times, ensuring a unique evaluation set for each iteration to prevent repeated assessment of slides within the internal dataset. For external evaluation, we utilized the LGNet-developed ensemble classifier model trained on the complete internal dataset to discern PCNSL from non-PCNSL. The prediction involved estimating the probability of classification for either PCNSL or glioma and for either PCNSL or non-PCNSL at the slide level. The outcomes obtained from both internal and external datasets were juxtaposed against the corresponding ground-truth tumor status. The details of these methodologies are provided in the Supplementary Methods section.

## Reader study

To evaluate the impact of the deep learning model on pathologists' diagnostic performance, we enlisted eight pathologists specializing in intraoperative neuropathological diagnosis. Pathologists 1–6 were affiliated with the Sun Yat-sen University Cancer Center, while Pathologists 7 and 8 were associated with Guangdong Provincial People's Hospital and The First Affiliated Hospital of Sun Yat-sen University. Their expertise levels varied: Pathologists 1 and 4 had one year of experience, Pathologists 2 and 5 had approximately 5 years, and Pathologists 3, 6, 7, and 8 had accrued up to 10 years of experience. Throughout the evaluation, these pathologists were kept blind to clinical dataset particulars, such as the PCNSL-to-glioma or PCNSL-to-non-PCNSL ratio, as well as the deep learning model's performance. For each WSI extracted from the external datasets, the pathologists executed dichotomous predictions indicating either PCNSL or glioma and either PCNSL or non-PCNSL. The process included an original diagnosis where the pathologist evaluated the slide independently, and a modified diagnosis following provision of the deep learning model's prediction. This included both predictive probability and binary classification for PCNSL or glioma, and PCNSL or non-PCNSL. Additionally, pathologists assigned self-confidence scores on a 6-scale for both original and modified diagnoses. For the classification of PCNSL or non-PCNSL, the scores ranged from '1' indicating 'surely non-PCNSL' to '6' signifying 'surely PCNSL'. The scores for PCNSL or glioma ranged similarly, representing the pathologists' confidence levels in their diagnoses. To better understand the association between specific histopathological characteristics and LGNet's predictions, we constructed a logistic regression model using frozen slides. The detailed description was shown in the supplementary methods.

## Human-machine fusion

To improve the diagnostic performance, we applied the human-machine fusion scheme, which is a simple extension of the fusion method originally developed in our previous study[27], further elaborated upon in the supplementary methods section. While the human-machine fusion strategy was implemented across all cases in the external datasets, only a select portion underwent this fusion method by pathologists in the proof-of-concept study.

Consequently, we referred to the fusion in the proof-of-concept study as the LGNet-Pathologist combination (L-P combination), distinguishing it from the human-machine fusion applied in the external cohorts.

## Evaluation of the model on the proof-of-concept study

To simulate real-world frozen diagnosis scenarios for pathologists with varying levels of experience, we conducted a proof-of-concept study using 68 frozen slides suspected of either PCNSL or glioma from our center. Two pathologists participated in the study: Pathologist A, with one year of experience in intraoperative neuropathological diagnosis, and Pathologist B, with 10 years of experience in intraoperative neuropathological diagnosis. Both pathologists were blinded to the primary intraoperative diagnosis and final postoperative diagnosis. To facilitate their decision-making process and visually display their decisions, we designed and developed an online pathological decision support platform accessible only to intranet users. The pathologists viewed the original and unprocessed H&E slides from the proof-of-concept study and made diagnoses based on their selected strategies, such as human-machine fusion or non-human-machine fusion. We compared the time spent by each pathologist from opening the frozen section to making the original diagnosis with that of LGNet's prediction. Furthermore, we compared the performance of LGNet with that of the two pathologists, as well as compared the performance of the LGNet-Pathologist combination, or human-machine fusion, with that of LGNet.

## Statistical analysis

The clinicopathological data in the retrospective cohorts were analyzed using Chi-square test or variance analysis. To compare the area under the receiver operating characteristic curves (AUROCs) between different variables, Delong's test was used. The cutoff threshold of deep learning model's ROC curve was determined by Youden's J statistic to dichotomize the model's probabilities into binary predictions. The McNemar test was used to compare the statistical differences in sensitivity and specificity. The association between LGNet prediction and morphological features was analyzed by logistic regression models. The Clopper-Pearson method was used to calculate 95% CIs. We considered a $P$ value less than 0.05 as statistically significant. The adjusted $P$ value with False Discovery Rate (FDR) method was also calculated when involving multiple testing. For statistical analysis, we used SPSS Statistics (version 20.0), Medcalc (version 15.2.2), and R (version 4.3.2). Python (version 3.9.6) and the deep learning platform PyTorch (version 1.9) were used for data preprocessing and model development. Some illustrations were generated with BioRender.com.

## Reporting summary

Further information on research design is available in the Nature Portfolio Reporting Summary linked to this article.

# Data availability

Due to patient privacy obligations and institutional regulations, restrictions are imposed on access to the whole-slide images and annotation data of both internal and external datasets used in this study. These datasets were obtained with institutional permissions via IRB (Institutional Review Board) approval and are therefore not publicly available. Nonetheless, for non-commercial and academic purposes, interested parties may request access to the data supporting the findings of this study directly from the corresponding author. Source data have been provided as a zip file with this paper. Source data are provided with this paper.

# Code availability

The source codes for our LGNet model have been made publicly accessible at https://github.com/Kepler1647b/LGNet/tree/main.

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

## Acknowledgements

This work was supported by grants from the National Key R&D Program of China (No. 2021YFA1300201, D.X.), the National Natural Science Foundation of China (grant No. 82172646, M.C.; 81872001, M.C.; 81972227, F.W. and 82073189, Y.W.), and the Guangdong Esophageal Cancer Institute Science and Technology Program (No.M202108, M.C.).

## Author contributions

M.C., D.X., Y.W. and Y.Y. conceived and designed the study. X. Zhang, P.L. and Z. Zhang collected the samples and acquired the image data. Y.Y., Y.W. and Q.C. provided the clinical and pathological data of multiple medical centers. Z. Zhao, R.W., H.C. and S.W. performed the machine learning. M.C., R.L., L.Liu, X. Zhang, L.Lan, X. Zheng, P.L., W.H., S.L. and Q.C. conducted the reader study. Z. Zhao and X. Zhang did the statistical analyses. All authors vouch for the data, analyses, and interpretations. X. Zhang, Z. Zhao, R.W., H.C., Y.Y., Y.W., D.X. and M.C. wrote the first draft of the manuscript, and all authors reviewed, contributed to, and approved the manuscript.

## Competing interests

The authors declare no competing interests.
