## [Peer Review File · Nature Communications]

A multicenter proof-of-concept study on deep learning-based intraoperative discrimination of primary central nervous system lymphomaEditorial Note: This manuscript has been previously reviewed at another journal that is not operating a transparent peer review scheme. This document only contains reviewer comments and rebuttal letters for versions considered at *Nature Communications*.

REVIEWER COMMENTS

Reviewer #1 (Remarks to the Author):

I would like to acknowledge that the authors have made considerable efforts in order to address the issues raised by the reviewers.

I still think that the presentation of some of the data remains biased:

- The finding that the processing time for the algorithm is shorter than the time pathologists need to come up with an interpretation is just besides point, as slide scanning time will be multiple times longer than either of the other steps (besides the problem of scanning freshly prepared frozen sections without drying them).
- In a similar way, the presentation of results in section 3 appears biased, in that lack of statistical significance is presented as "tendency" and issues with multiple testing are not addressed.

Reviewer #2 (Remarks to the Author):

The authors have addressed the issues I raised in my previous review and I have no further comments.

Reviewer #3 (Remarks to the Author):

The authors have addressed all of my comments. No further requests.

Reviewer #4 (Remarks to the Author):

Authors of this manuscript should be commended by their efforts to reply point by point to

the reviewers' comments. They were constructive and educational, discussing all the technical, methodological, interpretative, and applicative aspects of the study. The use of deep-learning machine (DLM) procedure to enhanced intraoperative differential diagnosis between PCNSL and other neurological disorders based on frozen sections has not yet been reported in the literature. However, it is important to put in context these new data to establish the potential applications of this study.

1) It is important to understand whether this is applicable to every patient with suspicion of PCNSL or only to patients with doubt diagnosis. Today, PCNSL suspicion is based on several parameters and not only on radiological features, site of disease and response to steroids as in the past. Other modern tools are useful to distinguish PCNSL from other CNS diseases, like radiomics [Guha, A., et al. Front Oncol 2022], genetic biomarkers [Ferrerri AJM, et al. Br J Haematol 2022] and cell-free DNA [Mutter JA, et al. J Clin Oncol 2022]. Authors did not use these modern tools to select doubtful cases for analysis, and this should be recognized as a limitation of the study. Moreover, regarding radiomics, authors state "the disparate outcomes of radiomics-based machine learning approaches in discerning PCNSL and glioma underscore their inconsistency and impede their clinical use", and cite ref #12. Ref #12 regards a meta-analysis concluding that "both deep learning and machine learning algorithms have demonstrated encouraging results and certainly have the potential to aid neurooncologists in taking preoperative decisions". In fact, that meta-analysis showed a sensitivity of 80-99%, specificity of 87-100% and a balanced accuracy of 82-100% in studies in which machine learning and deep learning were directly compared with the gold standard histopathology. Please correct this sentence in the introduction section.

2) Based on the analysis focused on "lesions categorized as ideally equivocal on preoperative imaging" (page 8), authors conclude "our data suggest that LGNet demonstrates robustness and consistent diagnostic performance". However, this seems overstated considering that 95%CI is ample, specially for specificity, with a range of more than 20%, and without achieving significant levels. Final and intermediate conclusions should be rewritten, avoiding overstatements.

3) PCNSL is an uncommon, aggressive but curable CNS tumor. To achieve this goal, patients should be managed by expert neurologists, neuroradiologist, neurosurgeons, neuropathologists, and hematooncologists. In referral centers, intraoperative pathological

exams are performed by expert neuropathologists, especially on difficult cases. Accordingly, the effect of the proposed DLM procedure in doubtful cases, assessed on small biopsies is of major interest. However, provided results are based on a few cases assessed by a few pathologists. This can be solved through two different strategies: 1) to enhance significantly (n= 15) the number of expert pathologists, or 2) to recognize that is only a proof-of-concept study that needs for future analyses on suitable number of pathologists and on better selected cases (small biopsies and difficult neuroimaging interpretation) to can draw reliable conclusions.

4) Some statements are incorrect other than overstated and should be rephrased correctly. Only for instance, in page 11, authors report “Additionally, concerning the external test datasets, the original fusion with pathologist 3 yielded an AUROC of 0.988 (95% CI: 0.98-1.00; P = 0.25), while the modified fusion reached an AUROC of 0.989 (95% CI: 0.98-1.00; P = 0.16), surpassing the model’s performance in distinguishing PCNSL from non-PCNSL (0.981)”. This is not correct as significant levels proposed by authors were not achieved, and should be interpreted as “no difference” or “a difference due to chance”. Please rephrase to avoid misinterpretation by general readers.

5) In line with the prior comment, I suggest rephrasing some overstated sentences in the discussion section. Only as an example, authors state “Surprisingly, pathologists assisted by LGNet, even with just one year of experience, demonstrated superior performance compared to their unassisted counterparts with five years of experience. Furthermore, their performance was on par with pathologists with ten years of experience” (page 16). All this part is overstated considering that authors compared results achieved on subgroups of 2-4 pathologists each. Please be more moderate with the weight of your conclusions; terms like “exceptional assistance” or “exceptional performance” should be replaced.

6) Based on the above-mentioned limitations is difficult to define the contribution of LGNet to enhance intraoperative diagnoses. Only for instance, LGNet exhibited misdiagnoses in 14% of PCNSL in External cohort 1 (page 14), which is not essentially different from the “ambiguous findings in a notable percentage of cases (approximately 10-20%)” reported in studies focused on intraoperative diagnosis and that authors cite in page 5. Please recognize this as an interpretation concern and limitation of the study in the discussion section.

Point-by-point response to the reviewers' comments:

Reviewer #1 (Remarks to the Author):

I would like to acknowledge that the authors have made considerable efforts in order to address the issues raised by the reviewers.

I still think that the presentation of some of the data remains biased:

- The finding that the processing time for the algorithm is shorter than the time pathologists need to come up with an interpretation is just besides point, as slide scanning time will be multiple times longer than either of the other steps (besides the problem of scanning freshly prepared frozen sections without drying them).

Our Reply:

Thank you for your careful reading and pointing this out. We value your perspective on the practical implications related to the processing time of our algorithm in comparison to the time required for pathologists to interpret slides. We recognize the importance of slide scanning time as a critical component of the intraoperative diagnostic workflow. In response to your feedback, we have opted to remove the relevant results from our revised manuscript.

- In a similar way, the presentation of results in section 3 appears biased, in that lack of statistical significance is presented as "tendency" and issues with multiple testing are not addressed.

Our Reply:

Thank you for your insightful feedback, which we greatly appreciate. We acknowledge the importance of meticulously presenting our results, particularly in section 3. In light of your comments, we have revised the presentation of results and employed adjusted p-values using the False Discovery Rate (FDR) method to mitigate any concerns related to multiple testing. Please refer to the revised section 3 and Tables 1, 3, as well as Supplementary Tables 6 to 8 for the updated analyses and findings.

Reviewer #2 (Remarks to the Author):

The authors have addressed the issues I raised in my previous review and I have no further comments.

Our Reply:

We thank the reviewer for the positive feedback.

Reviewer #3 (Remarks to the Author):

The authors have addressed all of my comments. No further requests.

Our Reply:

We are grateful to the reviewer's positive feedback.

Reviewer #4 (Remarks to the Author):

Authors of this manuscript should be commended by their efforts to reply point by point to the reviewers' comments. They were constructive and educational, discussing all the technical, methodological, interpretative, and applicative aspects of the study. The use of deep-learning machine (DLM) procedure to enhanced intraoperative differential diagnosis between PCNSL and other neurological disorders based on frozen sections has not yet been reported in the literature. However, it is important to put in context these new data to establish the potential applications of this study.

1) It is important to understand whether this is applicable to every patient with suspicion of PCNSL or only to patients with doubt diagnosis. Today, PCNSL suspicion is based on several parameters and not only on radiological features, site of disease and response to steroids as in the past. Other modern tools are useful to distinguish PCNSL from other CNS diseases, like radiomics [Guha, A., et al. Front Oncol 2022], genetic biomarkers [Ferreri AJM, et al. Br J Haematol 2022] and cell-free DNA [Mutter JA, et al. J Clin Oncol 2022]. Authors did not use these modern tools to select doubtful cases for analysis, and this should be recognized as a limitation of the study. Moreover, regarding radiomics, authors state “the disparate outcomes of radiomics-based machine learning approaches in discerning PCNSL and glioma underscore their inconsistency and impede their clinical use”, and cite ref #12. Ref #12 regards a meta-analysis concluding that “both deep learning and machine learning algorithms have demonstrated encouraging results and certainly have the potential to aid neurooncologists in taking preoperative decisions”. In fact, that meta-analysis showed a sensitivity of 80-99%, specificity of 87-100% and a balanced accuracy of 82-100% in studies in which machine learning and deep learning were directly compared with the gold standard histopathology. Please correct this sentence in the introduction section.

Our Reply:

Thank you for your thoughtful comments and for highlighting the importance of clarifying the applicability of our study findings to patients suspected of having PCNSL.

We agree that modern diagnostic tools, such as radiomics and genetic biomarkers, play a significant role in PCNSL diagnosis, and we acknowledge their potential utility in distinguishing PCNSL from other CNS lesions. We recognize that our study did not utilize these modern tools to select doubtful cases for analysis, and we appreciate your insight into this limitation. We have discussed this point in our revised manuscript accordingly.

Regarding the citation in the introduction section, we will carefully review the reference and revise the sentence accordingly to accurately reflect the findings of the meta-analysis cited. It now reads “Despite the promising results of radiomics-based machine learning approaches in discerning PCNSL and glioma, some models still exhibit room for improvement in performance, as highlighted in pooled analyses”.

2) Based on the analysis focused on “lesions categorized as ideally equivocal on preoperative imaging” (page 8), authors conclude “our data suggest that LGNet demonstrates robustness and consistent diagnostic performance”. However, this seems overstated considering that 95%CI is ample, specially for specificity, with a range of more than 20%, and without achieving significant levels. Final and intermediate conclusions should be rewritten, avoiding overstatements.

Our Reply:

Thank you for your insightful comment. In response to the reviewer's advice, we have revised several conclusions in the updated manuscript. Please refer to the corresponding sections in our revised manuscript highlighted in red ink for your convenience.

3) PCNSL is an uncommon, aggressive but curable CNS tumor. To achieve this goal, patients should be managed by expert neurologists, neuroradiologist, neurosurgeons, neuropathologists, and hematooncologists. In referral centers, intraoperative pathological exams are performed by expert neuropathologists, especially on difficult cases. Accordingly, the effect of the proposed DLM procedure in doubtful cases, assessed on small biopsies is of major interest. However, provided results are based on

a few cases assessed by a few pathologists. This can be solved through two different strategies: 1) to enhance significantly (n= 15) the number of expert pathologists, or 2) to recognize that is only a proof-of-concept study that needs for future analyses on suitable number of pathologists and on better selected cases (small biopsies and difficult neuroimaging interpretation) to can draw reliable conclusions.

Our Reply:

Thank you for your valuable insights. We acknowledge the importance of assessing the effect of the proposed DLM procedure in doubtful cases, particularly those involving small biopsies, in the management of PCNSL. As you rightly pointed out, our study is a proof-of-concept study, and we recognize the need for future analyses involving a larger number of expert pathologists and better-selected cases to draw more reliable conclusions. In our revised manuscript, we have highlighted this limitation and emphasized the importance of further investigation in this direction to enhance the clinical applicability and robustness of our findings (please refer to the discussion section).

4) Some statements are incorrect other than overstated and should be rephrased correctly. Only for instance, in page 11, authors report “Additionally, concerning the external test datasets, the original fusion with pathologist 3 yielded an AUROC of 0.988 (95% CI: 0.98-1.00; P = 0.25), while the modified fusion reached an AUROC of 0.989 (95% CI: 0.98-1.00; P = 0.16), surpassing the model’s performance in distinguishing PCNSL from non-PCNSL (0.981)”. This is not correct as significant levels proposed by authors were not achieved, and should be interpreted as “no difference” or “a difference due to chance”. Please rephrase to avoid misinterpretation by general readers.

Our Reply:

Thank you for bringing this to our attention. We appreciate your diligence in scrutinizing our manuscript for accuracy. We have rephrased the statement to accurately reflect that the difference observed in the performance metrics between the original fusion and modified fusion models did not reach statistical significance, indicating that there was no significant difference or that any observed difference may

have occurred by chance. We ensure that the revised statement conveys the appropriate level of certainty and avoids overstating the findings in our updated manuscript (please refer to the multiple places of the result section).

5) In line with the prior comment, I suggest rephrasing some overstated sentences in the discussion section. Only as an example, authors state “Surprisingly, pathologists assisted by LGNet, even with just one year of experience, demonstrated superior performance compared to their unassisted counterparts with five years of experience. Furthermore, their performance was on par with pathologists with ten years of experience” (page 16). All this part is overstated considering that authors compared results achieved on subgroups of 2-4 pathologists each. Please be more moderate with the weight of your conclusions; terms like “exceptional assistance” or “exceptional performance” should be replaced.

Our Reply:

As requested, we’ve rephrased some overstated sentences in the discussion section of our revised manuscript.

6) Based on the above-mentioned limitations is difficult to define the contribution of LGNet to enhance intraoperative diagnoses. Only for instance, LGNet exhibited misdiagnoses in 14% of PCNSL in External cohort 1 (page 14), which is not essentially different from the “ambiguous findings in a notable percentage of cases (approximately 10-20%)” reported in studies focused on intraoperative diagnosis and that authors cite in page 5. Please recognize this as an interpretation concern and limitation of the study in the discussion section.

Our Reply:

We appreciate the reviewer's thorough review and insightful comments. The observed misdiagnosis rate of LGNet is comparable to the misdiagnosis rate reported in previous studies for pathologists, which typically falls within the range of approximately 10-20%. In our investigation, LGNet demonstrated a misdiagnosis rate of 14% (7/49) for PCNSL cases in External Cohort 1. Notably, all instances of misdiagnosis by LGNet were

correctly identified by pathologists with five or ten years of experience in intraoperative neuropathological diagnosis. This observation suggests a complementary relationship between pathologists and the model in the intraoperative diagnosis of PCNSL and underscores the potential for enhancing diagnostic accuracy by integrating deep learning algorithms with pathologists' expertise. We have acknowledged this aspect and included it in the discussion section of the manuscript.

REVIEWERS' COMMENTS

Reviewer #1 (Remarks to the Author):

The authors have again made a considerable effort to address the issues raised. I have no further comments.

Reviewer #4 (Remarks to the Author):

The authors have addressed the issues I raised in my previous review and I have no further comments.